# G870A Polymorphic Variants of *CCND1* Gene and Cyclin D1 Protein Expression as Prognostic Markers in Laryngeal Lesions

**DOI:** 10.3390/diagnostics12051059

**Published:** 2022-04-23

**Authors:** Magdalena Marianna Kowalczyk, Magda Barańska, Wojciech Fendler, Edyta M. Borkowska, Józef Kobos, Maciej Borowiec, Wioletta Pietruszewska

**Affiliations:** 1Department of Otolaryngology, Head and Neck Oncology, Medical University of Lodz, 90-153 Lodz, Poland; mk6.kow@gmail.com (M.M.K.); magda.a.baranska@gmail.com (M.B.); 2Department of Radiation Oncology, Dana-Farber Cancer Institute, Boston, MA 02115, USA; wojciech.fendler@umed.lodz.pl; 3Department of Biostatistics and Translational Medicine, Medical University of Lodz, 90-419 Lodz, Poland; 4Department of Clinical Genetics, Medical University of Lodz, 92-213 Lodz, Poland; edyta.borkowska@umed.lodz.pl (E.M.B.); maciej.borowiec@umed.lodz.pl (M.B.); 5Department of Pathology, Medical University of Lodz, 92-215 Lodz, Poland; jozef.kobos@umed.lodz.pl

**Keywords:** larynx cancer, precancerous laryngeal lesions, *CCND1* gene, Cyclin D1, prognostic factors

## Abstract

*CCND1* gene encodes Cyclin D1 protein, the alternations and overexpression of which are commonly observed in human cancers. Cyclin D1 controls G1-S transition in the cell cycle. The aim of the study was to assess utility of the genotyping and protein expression in predicting the susceptibility of transformation from normal tissue to precancerous laryngeal lesions (PLLs) and finally to laryngeal cancer (LC). Four hundred and thirty-five patients (101 with LC, 100 with PLLs and 234 healthy volunteers) were enrolled in the study. Cyclin D1 expression was examined by immunohistochemistry and G870A polymorphism of gene *CCND1* by PCR-RFLP technique. We confirmed association between the A allele and risk of developing LC from healthy mucosa (*p* = 0.006). Significantly higher expression of Cyclin D1 was observed in LC compering with PLLs (*p* < 0.0001) and we found that it could be a predictive marker of shorter survival time. To sum up, in the study population *CCND1* gene polymorphism A870G and Cyclin D1 expression have a significant impact on the risk of developing PLLs and LC, and, therefore, Cyclin D1 could be a useful marker for the prediction of survival time in LC, whereas *CCND1* gene polymorphism does not have a direct impact on patients’ outcome.

## 1. Introduction

Laryngeal cancer (LC) stands for more than a third of head and neck cancers, which makes it one of the most common neoplasms in this anatomic region. About 85–90% of all LCs are laryngeal squamous cell carcinomas. Etiology is multifactorial, both the genetic and environmental factors participate in the development of the disease, well-known risk factors are tobacco and alcohol. For many years, the five-year survival rate has remained low: 63.2% in 1975 and 60.4% in 2010, despite the development of new diagnostic and therapeutic techniques [1,2]. About 90% of laryngeal malignant tumors of the larynx arise from premalignant lesions [3]. Those are defined as morphological alterations of the mucosa caused by irritative factors or generalized illness [4]. Premalignant laryngeal lesions (PLLs) generally have the same symptoms as early LC. Then, it is clinically impossible to distinguish PLL from LC in early stage using current methods [5]. PLLs, according to the WHO classification from 2017, are histologically classified into low-grade and high-grade dysplasia. This two-tier classification can be transformed into a three-tier system with a distinction between high-grade dysplasia and carcinoma in situ [6]. Conventional prognostic factors such as TNM staging, grading and localization of lesion are inadequate in patients’ outcome prediction. Additionally, early diagnosis of laryngeal cancer remains a challenge. For this reason, new methods of endoscopic examination are being developed. One of the first was staining with toluidine blue or Lugol’s solution, but they gave a high percentage of false positive diagnoses and required general anesthesia for direct laryngoscopy [7,8]. In recent years, biological endoscopic evaluation enables the visualization of neoangiogenesis and makes the larynx more accessible for examination and monitoring of lesions. Both narrow-band imaging and Storz SPIES are in use, which allow the detection of lesions smaller than 5 mm and carcinomas in situ. Our study and others have previously shown the high usefulness of recently used methods of biological endoscopy, especially NBI, in diagnosing oropharyngeal and laryngeal pathological lesions [9,10].

Despite the availability of new diagnostic techniques and well-established histopathological analysis in the clinical setting we cannot foresee the disease outcome and optimal therapy. This provides a rationale for searching molecular biomarkers, which recently demonstrated practical utility [11]. Much effort is currently being made to explore the molecular landscape of head and neck cancers. One of the most frequent changes in the head and neck squamous cell carcinomas in 17.6–37% are alterations in the *CCND1* gene [12]. It is located on chromosome 11q13 and its aberrations such as amplifications, translocations and inversions have been found frequently in human tumors such as head and neck cancer [13], non-small cell lung cancer and esophageal adenocarcinoma [14]. Its product, Cyclin D1 is one of the key role proteins in the regulation of the cell cycle and response to mitotic factors forming complexes with CDK4 and CDK6 leading to phosphorylation of the RB protein. Inhibition of Cyclin D1 causes cell cycle arrest, whereas overexpression of the protein accelerates the G1 phase transition and is believed to be connected with tumorigenesis [15]. Therefore, increased expression of Cyclin D1 plays a decisive role in tumor formation and the maintaining of the malignant phenotype [16]. Overexpression of Cyclin D1 was observed in lung, mouth, throat, larynx, esophagus and colorectal cancer [17,18,19,20,21,22]. In advanced larynx cancer, overexpression of Cyclin D1 is a known predictive marker of both overall and disease-free survival. Furthermore, the authors claim that adding the Cyclin D1 to the traditional staging system has the potential to enhance it [23].

Common single nucleotide polymorphism (SNP) in the *CCND1* gene (rs9344) G870A is associated with the increased risk of cancer [24,25,26]. This polymorphism is based on the G→A transition of the last nucleotide of exon 4 and the conserved mRNA assembly site at the interface between exon 4 and intron 4 [27]. The consequence of the replacement is alternative gene splicing, which means that the alternative transcript has no exon 5. Polymorphic variants of the G870A produce alternative transcripts with different half-lives. The *CCND1* 870A variant preferentially encodes the altered transcript b (cyclin D1b), even in the heterozygous state [28]. Cyclin D1b has a five-fold longer nuclear half-life than its wild form transcript a (cyclin D1a) [29]. Expression of both isoforms is elevated in multiple neoplasms and both play an important role in tumorigenesis, so there was no need to separately measure Cyclin D1a and Cyclin D1b to assess risk of cancer transformation. However, studies revealed that Cyclin D1b seems to be a stronger oncogene, as it could transform cells more easily compared to Cyclin D1a [30]. Alteration of both the *CCND1* gene and Cyclin D1 protein are frequently found in precancerous lesions and neoplasms. There are inconsistencies if and how they affect PLLs appearance and progression to LC.

Our purpose was to investigate the utility of the *CCND1* genotyping and Cyclin D1 expression in predicting susceptibility of transformation from normal tissue to precancerous laryngeal lesions (PLLs) and to laryngeal cancer (LC).

## 2. Materials and Methods

### 2.1. Study Population and Epidemiologic Data

The study group was recruited prospectively from patients surgically treated by choice in the Department of Otolaryngology, Head and Neck Oncology, Medical University of Lodz from 2005–2010. Inclusion criteria were adult participants with histologically confirmed laryngeal cancer or any premalignant laryngeal lesion who gave informed consent for surgery and who agreed to participate. Postsurgical surveillance was performed every 1 month during the first year after surgery, every 3 months for two additional years, then every 6 months and once a year after 5 years of follow up. The total follow-up time was almost 12 years in patients with laryngeal cancer and 15 years in patients with precancerous lesions. The samples from the PLL with progression to LC group were taken and analyzed at the beginning when they were diagnosed as the premalignant lesions. We did not collect another sample when the lesion progressed to cancer during the follow-up.

Healthy controls, Caucasians from the same residential area, were recruited from the patients surgically treated in our department due to a non-oncological purpose who had no previous history of cancer and immunological disorders including septoplasty, ossiculoplasty, myringoplasty and trauma injuries. Control cases were matched and similarly adjusted to age (±5 years), gender, smoking and alcohol consumption habits. Only blood samples for the genotyping were obtained from controls.

Tobacco exposure was expressed as pack-years of smoking. We distinguished non-smokers and smokers, among whom we differentiated light and heavy, respectively, below and above the median (7300 cigarettes per year), due to the same value in all groups. We assessed only the data about active smoking. Patients due to alcohol consumption were grouped into non-drinkers and those who drink less or more than six beverages weekly (median).

We evaluated all items from the REMARK checklist for tumor marker prognostic studies and followed them (Appendix A) [31].

The research was approved by the Local Ethics Committee (RNN/12/06/KE).

All procedures involving human subjects were performed according to the Helsinki Declaration. Informed consent was obtained from all patients prior to study inclusion.

### 2.2. Immunohistochemistry

The appropriate tissue samples were selected from laryngeal squamous cell carcinoma and from premalignant laryngeal lesions independently. Tissue samples were selected from paraffin-embedded blocks and cut into 4-m thick sections. They were deparaffinized in xylene and dehydrated through graded alcohols. Endogenous peroxidase was blocked with 3% H2O2 after 5 min incubation (pressure cooking) in 10 mmol/L of citrate buffer (pH 6.0) or EDTA buffer (pH 9.0). The next step was 20 min incubation with the serum blocking solution. The real EnVision detection system (DAKO) was used for immunostaining (according to the manufacturer’s procedure). The mouse anti-Cyclin D1 monoclonal antibody (NCL-Cyclin D1, clone designation DCS-6, Novocastra Laboratories Ltd., Newcastle upon Tyne, UK) was used to detect total Cyclin D1. Next was 2 h incubation of primary antibody at room temperature, then washing (3 times) with phosphate-buffered saline, 30 min incubation using labeled polymer-horseradish peroxidase (HRP) antimouse/rabbit Ig. The diaminobenzidine chromogen solution was added after incubation. Counterstaining was conducted with hematoxylin. Negative controls were performed by omitting the primary antibody. The nuclear staining was considered positive, regardless of cytoplasmic staining. The morphometric evaluation was made using a light microscope equipped with a special program (Nikon, magnification 200 and 400×). A positive reaction was recognized as brown coloration of the cell nuclei and determined from the staining intensity as well as the percentage of immunoreactive cells in the hot spots (most highly stained area of each slide) based on a previously reported method (the index of positive cells). The intensity score was graded as 0 (no staining), 1 (weak staining), 2 (moderate staining) and 3 (severe staining).

Furthermore, in PLLs, a quantitative indication of staining was determined by calculating the percentage of immunopositive epithelial cells to the total epithelial cells, to create an expression index (EI). Sections of the cervix with different grades of intraepithelial neoplasia were used as positive controls. Staining intensity was not included in the assessment, as only relatively minor variations in staining intensity were observed.

The results were independently assessed by specialists in histopathology. Cases with divergent results were re-evaluated simultaneously and a final opinion was made and based on consensus by both the investigators.

### 2.3. Genotyping

Blood samples (30 mL) from patients with laryngeal lesions and healthy volunteers were collected in coded, EDTA-containing tubes that were sent to the laboratory for immediate DNA extraction. Laboratory personnel were blinded to case-control status. Genomic DNA was isolated from peripheral blood lymphocytes using the DNA Qiagen Blood Mini Kit (Qiagen, Valencia, CA, USA) according to the manufacturer’s protocol.

Sequences for genotyping probes and primers were either obtained from the SNP500 Cancer database or designed using the PrimerExpress 2.0 software (Applied Biosystems, Waltham, MA, USA).

The CCND1 G870A (exon 4—intron 4 boundary) polymorphism was detected by the PCR-RFLP method. The 167-bp fragment of the CCND1 gene at the junction of ex-on 4/intron 4, containing polymorphic nucleotide, was amplified in Eppendorf Mastercycler gradient thermal cycler (Hamburg, Germany) using two primers (forward 5′-GTGAAGTTCATTTCCAATCCGC-3′; reverse 5′-GGGACATCACCCTCACTTAC-3′; Life Technologies, Inc.-Invitrogenin) a total reaction volume of 30 µL containing 100 ng of genomic DNA, 1 U Taq DNA polymerase (Sigma, Burlington, MA, USA), 1·PCR buffer (10 mM Tris-HCl, pH 8.3, 50 mM KCl, 1.5 mM MgCl2), 0.2 mM of each dNTP (Roche, Germany), denaturation for 10 min at 94 °C, 35 cycles of 30 s 94 °C, 30 s 60 °C, 1 min at 72 °C and terminal extension for 10 min at 72 °C. PCR products were analyzed in 1.8% agarose gel and digested overnight (5 U of Scr FI restriction endonuclease, New England Biolabs Inc., Beverly, MA, USA) at 37 °C followed by electrophoresis in 2.5% agarose gel. Five percent of all samples were randomly selected and genotyped in duplicate with 100% concordance. The analysis was conducted for two genotypes (combined AA/AG and GG form) due to the dominant biological functional effect of the A allele (minor allele) on half-life Cyclin D1 protein.

### 2.4. Statistical Analysis

Continuous variables were presented as medians with interquartile ranges (Me (IQR)) due to non-normal distribution. Nominal variables were presented as percentages. Comparisons were performed using U Mann–Whitney tests for continuous variables and Yates’ corrected Chi2 test or Fisher’s exact test for nominal variables depending on the number of analyzed samples. The PLLs and LC associations with the evaluated genotype were calculated as odds ratios (OR) and 95% confidence intervals (95%CI) using unconditional multivariate logistic regression adjusted for age, gender, primary tumor localization, T characteristic, nodal involvement, TNM status, histological grading, smoking and alcoholic status (never, light and heavy).

In patients with PLLs with progression to cancer and all LC cases, receiver operating characteristic (ROC) curves were constructed to determine cut-off values for tissue Cyclin D1 protein expression in PLLs for overall and disease-free survival analysis. Studies of time-dependent outcomes were performed using Kaplan-Meier curve analysis and univariate Cox proportional hazard model. Factors with a *p*-value of <0.15 in univariate comparisons were entered into a backward stepwise multivariate analysis retaining factors with a *p*-value of <0.15. A *p*-value of <0.05 was considered statistically significant in final analyses. Statistica 12.0 (StatSoft, Tulsa, OK, USA) and Medcalc 9.3 (Medcalc, Mariekerke, Ostend, Belgium) software were used.

## 3. Results

### 3.1. Patients’ Characteristics

Four hundred and thirty-five patients (100 with PLLs, 101 with LC and 234 healthy volunteers) were enrolled in the study. The study group (n = 201) consisted of three patient subgroups: laryngeal cancer (n = 101), premalignant laryngeal lesions (n = 83) and PLLs with progression to cancer (n = 17). All cancer cases were histologically confirmed as squamous cell carcinomas. After the surgery, patients with PLLs were observed for up to 15.3 years (mean 13.1, SD 2.2; median 13.0). In this group of patients, progression to laryngeal cancer was observed in the period from 8.5 to 12.8 years (mean 10.3; SD 2.5; median 11.0). The observation period of patients with laryngeal cancer ranged from 1 to 11.2 years after the diagnosis and surgical treatment of laryngeal cancer (mean 6.0; cf. SD 3; median 6.0). Three-year survival in this group was found in 72 patients (71.3%) and five-year survival in 63 patients (62.4%). In 27 patients with laryngeal cancer (26.7%), we observed local or nodal recurrences, which appeared in a time range between 12 months and 8.7 years (mean 23; BC 25.13; median 13). The control group involved 234 healthy patients. The demographic and clinical data are described in Table 1. In all further analyses, patients were divided into two groups, ≤55 and >55 years, based on the median age in PLLs and LC groups. Premalignant lesions were histologically graded according to the WHO 2005 and re-evaluated with 2017 WHO guidelines [6,32].

In a further step, CCND1 G870A polymorphism distribution and Cyclin D1 expression in all three groups of patients with laryngeal lesions were analyzed. Risk factors for the development of a precancerous lesion and laryngeal cancer were identified. Furthermore, clinical and genetic factors associated with overall and disease-free survival of patients with laryngeal cancer were analyzed.

### 3.2. CCND1 G870A Polymorphism Distribution in Patients with Laryngeal Lesions

In the controls, observed allele and genotype frequencies did not show significant deviation from the Hardy-Weinberg principle (χ^2^, two degrees of freedom, HW-E *p* = 0.58, Table 2). The frequency of the GG genotype was higher in controls than in all examined group with laryngeal lesions: the PLLs, PLLs with progression and LC patients (27.3%, 24.1%, 11.8% and 12.9% respectively). In patients with laryngeal cancer genotypes AA and AG were more common in poorly differentiated tumors (G3) (*p* = 0.04). In all study groups, there were no additional statistically significant differences in the distribution of CCND1 genotype and other clinical and histological factors mentioned in Table 1. The equations should be inserted in editable format from the equation editor.

### 3.3. Cyclin D1 Expression in Patients with Laryngeal Lesions

Expression of Cyclin D1 protein was found in 76 (91.6%) PLLs, 16 (94.1%) PLLs with progression to LC and 98 (97.0%) LC. Cyclin D1 was expressed in the basal and middle third of the epithelium in the low grade, and with full-thickness expression in high-grade dysplasia and carcinoma in situ (Figure 1). Cyclin D1 expression differed significantly between PLLs (median: 10 (5–15)) and LC (median: 20 (10–35)) (*p* < 0.0001). Moreover, in PLLs with progression, initial expression of Cyclin D1 were lower than in the cancer tissue during the follow-up (median: 5 (IQR 1–10) and 26 (IQR 10–40), *p* = 0.002). Expression of Cyclin D1 increased along with progression from low-grade (median—10%), through high-grade dysplasia (15%) to carcinoma (30%) (*p* = 0.01).

Cyclin D1 protein expression was not associated with G/A polymorphic variants distribution in LC (median: 25 in GG group (IQR 15–35) and 20 (IQR 10–35) in AA/AG group; *p* = 0.48). Conversely, in PLLs patients, protein tissue levels were higher in A allele carriers (median: 5 in GG group (IQR 2–10) and 10 (IQR 5–20) in AA/AG group; *p* = 0.06) but the difference was only close to statistical significance. Other clinical data mentioned in Table 3 were not associated with Cyclin D1 expression.

### 3.4. Analysis of Risk Factors for Laryngeal Lesions Development

Males and patients older than 55 years old were more predisposed to both PLLs and LC in univariate and multivariate analysis. Assessing the risk for LC, alcohol consumption retained its significance in the univariate analysis only, but smoking was also an important risk factor in multivariate analysis. Epiglottic localization, high expression of Cyclin D1 and male gender increased the risk of progression from PLLs to LC in univariate analysis, but the difference did not reach the level of significance. Patients with high-grade dysplasia were more predisposed for progression compared to low-grade but the difference did not reach the level of significance (Table 4). Allele A increased the odds of LC approximately three-fold. In contrast, the distribution of genotypes was not associated with the occurrence of a precancerous laryngeal lesions or its progression to laryngeal cancer.

### 3.5. Analysis of Overall and Disease-Free Survival of Patients with Laryngeal Cancer

Male sex, epiglottic localization of the tumor, metastases in the neck lymph nodes, high primary tumor extension (T3, T4), high histological grading (G3) and high expression of the Cyclin D1 protein (above the median) were significantly associated with shorter survival time of patients with larynx cancer in univariate analysis. Moreover, patients with Cyclin D1 expression index above 15% (a cut-off point) were three times more likely to die from laryngeal cancer (Appendix A). The CCND1 G870 A polymorphism was not statistically significant in the univariate analysis of overall survival.

In multivariate analysis, well known clinical prognostic factors were associated with the risk of death from LC. The presence of neck lymph node metastases increased the likelihood of death due to cancer by over five times; high histological grading (G3) resulting in increased risk of death over 2.5-fold and also high expression of the Cyclin D1 protein (above the median). Considering the cut-off point determined at the level of 15% using ROC curves, it was found that the relationship between high Cyclin D1 protein expression and shorter survival time of larynx cancer patients was stronger compared to the median threshold and the risk of dying from cancer increased 2.5-fold (Appendix A). Factors associated with overall survival are summarized in Table 5.

Disease-free survival of patients with laryngeal cancer in univariate analysis was related to nodal involvement (OR 3.7; 95%CI 1.71–8.04; *p* < 0.001), high histological grading (G3) (OR 1.77; 95% CI 1.1–3.14; *p* = 0.04) and epiglottic localization of primary tumor (OR 3.25; 95%CI 1.98–5.43; *p* < 0.001). In multivariate analysis, nodal involvement was the only indicator of death due to relapse of larynx cancer and increased the risk of recurrence almost four times (HR 3.9; 95% CI 1.89–8.24; *p* = 0.0009; for the whole model chi^2^ = 36.7; *p* < 0.0001). Allele A CCND1 G870A polymorphism carriers and Cyclin D1 protein overexpression did not affect laryngeal cancer recurrence.

## 4. Discussion

Obtained results show that the expression of Cyclin D1 has an important role in the progression of premalignant lesions to laryngeal cancer. Additionally, the common polymorphic variant A870 in the CCND1 gene may be a risk factor of cancer transformation. The wild-type GG genotype was more common in controls and seemed to be a protective variant decreasing the risk of LC development. It was also confirmed in our study that the presence of A allele, which increases the risk of tumorigenesis, was more frequent in PLLs and LC. To our knowledge, this is the first research that comprehensively explores the association of CCND1 polymorphism and Cyclin D1 protein expression with risk for both PLLs and LC, including cases of PLLs with progression to cancer in the longitudinal study. Moreover, there is only one paper concerning this polymorphism in Polish patients with laryngeal cancer, but the study group was much smaller and did not include premalignant lesions (n = 63 LC and n = 102 controls) [33]. Considering that some relations in one ethnic group may not work in another, we decided to extend the scope of this research in the Polish population. Research conducted in US populations revealed comparable results to ours. Marsit el al. showed a higher probability of oral and laryngeal cancer in comparison to other head and neck locations for A allele carriers compared to GG homozygotes [34]. Nishimoto et al. observed a significant increase in the risk of developing upper respiratory tract and gastrointestinal cancer in non-drinkers and non-smokers of the A allele carriers, but with boundary significance [20]. Zheng et al. also indicated a high risk of developing head and neck cancer in non-smoking, non-drinking patients under 50 years of age and in women who are homozygous AA [24]. This suggests that genetic factors could be strong indicators of predisposition to cancer and in some cases even stronger than environmental factors. However, some studies have revealed that GG homozygotes were associated with quicker relapse, shorter disease-free interval and reduced overall survival in LC patients at high risk for oral squamous cell carcinoma [33,35,36,37,38]. Our results concerning allele A as the factor increasing the risk of the LC are consistent with recently published meta-analysis investigating the association between the G870A CCND1 polymorphism and cancer in the upper aerodigestive tract. The study also suggests that homozygous AA status is associated with longer disease-free intervals and the low grade [39]. However, that study was performed in an ethnically mixed population, which might be the reason for the dissimilarity with the results of other researchers.

All these studies focus on LC. We found only two publications considering CCND1 G870A polymorphism and Cyclin D1 expression in PLLs but performed in small and ethnically mixed groups. Furthermore, authors focus mainly on response to combination biochemopreventive therapy (13-cis-retinoic acid, α-interferon and α-tocopherol) [19,40].

Moreover, in our study, among patients with premalignant laryngeal lesions, A allele carriers of the CCND1 gene G870A polymorphism had higher expression of Cyclin D1 protein compared to GG homozygotes. It was observed in patients with PLLs that progressed to cancer as well. This trend was also visible in high-grade dysplasia cases compared to the patients with early stages of PLLs. Our results showed that expression of Cyclin D1 increased on progression along with the lesion—from dysplasia (10–15%) to carcinoma (30%). Marzic et al. obtained a similar pattern of expression but with different values—35% for premalignant lesions and 45% for cancer [41]. These results suggest that cyclin D1 expression can be used for differentiating the PLLs from cancer but not from the healthy tissue. However, some researchers showed that Cyclin D1 appeared to be highly sensitive (81.2%) and a specific marker (83.9%) not only in differentiating LC from PLLs but also from healthy laryngeal mucosa (sensitivity: 81.2%; specificity: 41.4%) [42]. Recent studies also revealed that the expression of Cyclin D1 protein was correlated with tumor stage, metabolic volume of tumor, glycolysis in lesion and maximum standardized uptake value (SUV max) in patients diagnosed with laryngeal cancer and was suggested as a diagnostic and follow-up marker [34,43]. Moreover, studies showed that overexpression of Cyclin D1 is associated with lymph node metastases (OR 2.26; 95% CI 1.61–3.16) [44]. We did not observe an association with tumor extension, nodal involvement, TNM stage and histological grading of LC. Our results show that Cyclin D1 protein expression was significantly associated with a shorter survival time in laryngeal cancer patients, which is similar to the results of other studies [23].

## Figures and Tables

**Figure 1 diagnostics-12-01059-f001:**
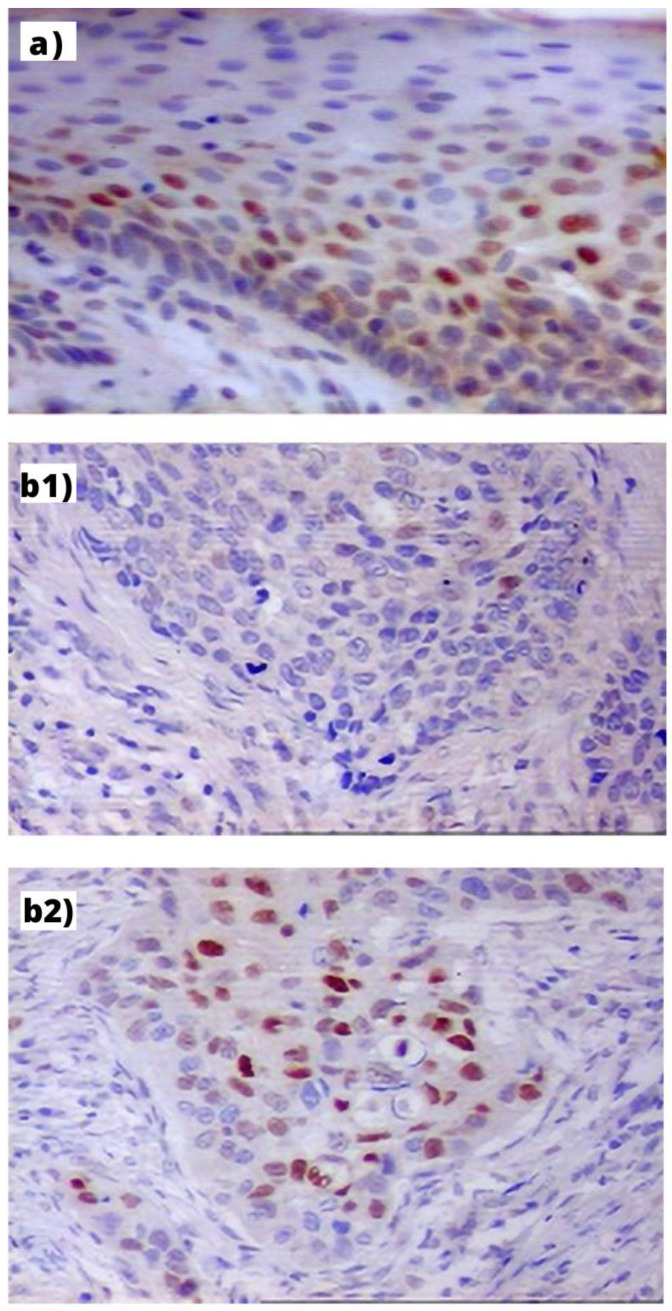
Immunohistochemical expression of Cyclin D1 in: (**a**) PLL—visible mostly in the nuclei of basal layer cells of hyperplastic epithelium (index: 32%, orig. magn. 200×); (**b**) LC: (**b1**) in the GG homozygote patient—visible in single cancer cells (index: 5%, orig. magn. 200×); (**b2**) in the AA homozygote patient visible as small cellular granules in nuclei of cells (index: 41%, orig. magn. 200×).

**Table 1 diagnostics-12-01059-t001:** Characteristics of examined groups of patients with premalignant laryngeal lesions (PLLs) and laryngeal cancer (LC) (PLLs→LC—PLLs with progression to LC).

	Control (n = 234)	PLLs (n = 83)	PLLs→LC (n = 17)	LC (n = 101)
**Mean age (range)**	47.4 (38–76)	52.0 (20–76)	54.8 (37–76)	57.4 (37–82)
**Gender**				
**Male**	123 (52.3%)	56 (67.5%)	16 (94.1%)	91 (90.1%)
**Female**	111 (47.7%)	27 (32.5%)	1 (5.9%)	10 (9.9%)
**Smoking**				
**Non-smoker**	60 (25.6%)	18 (21.7%)	2 (11.8%)	6 (5.9%)
**light**	58 (24.8%)	16 (19.3%)	4 (23.5%)	19 (18.8%)
**heavy**	116 (49.6%)	49 (59%)	11 (64.7%)	76 (75.3%)
**Alcohol intake**				
**Non-drinker**	140 (59.8%)	42 (50.6%)	2 (11.8%)	9 (8.9%)
**light**	70 (29.9%)	29 (34.9%)	11 (64.7%)	72 (71.3%)
**heavy**	24 (10.3%)	12 (14.5%)	4 (23.5%)	20 (19.8%)
**Primary site**				
**epiglottic**		21 (25.3%)	5 (29.4%)	45 (44.6%)
**glottic**		61 (73.5%)	12 (70.6%)	53 (52.5%)
**subglottic**		1 (1.2%)	0	3 (2.9%)
**Tumor extension**				
**T1, T2**			5 (29.4%)	33 (32.7%)
**T3, T4**			12 (70.6%)	68 (67.3%)
**Nodal involvement**				
**N0**			14 (82.3%)	65 (64.4%)
**N1, N2, N3**			3 (17.7%)	36 (35.6%)
**TNM stage**				
**I**			4 (23.5%)	15 (14.8%)
**II**			1 (5.9%)	18 (17.8%)
**III**			9 (52.9%)	24 (23.8%)
**IV**			3 (17.7%)	44 (43.6%)
**Histological grading**				
**G1**			10 (58.8%)	25 (24.8%)
**G2**			6 (35.3%)	56 (55.4%)
**G3**			1 (5.9%)	20 (19.8%)
**Dysplasia grading**				
**low-grade**		54 (65%)	14 (82.3%)	
**high-grade**		29 (35%)	3 (17.7%)	
**Reccurence**				
**local**				10 (9.9%)
**nodal**				18 (17.8%)

**Table 2 diagnostics-12-01059-t002:** Distribution of the CCND1 G870A polymorphism genotypes (* reference group).

CCND1 Genotype	Control (n = 234)	PLLs (n = 83)	PLLs→LC (n = 17)	LC (n = 101)	All Study Groups vs. Control (OR 95% CI)	LC vs. Control (OR 95% CI)
**AA**	52 (22.2%)	33 (39.8%)	7 (41.2%)	26 (25.7%)	1.42 (0.98–2.21) *p* = 0.04	1.07 (0.68–1.54) *p* > 0.05
**AG**	118 (50.5%)	30 (36.1%)	8 (47.0%)	62 (61.4%)		
**GG**	64 (27.3%)	20 (24.1%)	2 (11.8%)	13 (12.9%)	1 *	1 *
**A allele carrier (GA/AA)**	170 (72.7%)	63 (75.9%)	15 (88.2%)	88 (87.1%)	1.72 (1.07–2.77) *p* = 0.009	2.55 (1.33–2.9) *p* = 0.006

**Table 3 diagnostics-12-01059-t003:** Expression of Cyclin D1 due to clinical, histological and epidemiological criteria in patients with laryngeal lesions (IA—impossible analysis).

Cyclin D1 Median (IQR)	PLLs (n = 83)	PLLs→LC (n = 17)	LC (n = 101)
**Age**			
**<55**	10 (5–20)	30 (5–40)	20 (5–35)
**>55**	10 (5–10)	20.5 (12.5–35)	20 (10–40)
** *p* **	0.94	0.82	0.73
**Gender**			
**Female**	10 (2–15)	5 (5–5)	10 (5–20)
**Male**	10 (5–15)	28 (12–40)	20 (10–35)
** *p* **	0.73	0.001	0.055
**Primary site**			
**Epiglottic**	10 (7.5–15)	26 (15–30)	25 (10–35)
**Glottic**	10 (5–15)	22.5 (10–43)	20 (10–35)
** *p* **	0.23	0.84	0.43
**Tumor extension**			
**T1, T2**		40 (15–46)	15 (10–30)
**T3, T4**		20.5 (7.5–32.5)	20 (10–35)
** *p* **		0.15	0.35
**Nodal involvement**			
**N0**		22.5 (10–40)	20 (10–35)
**N1, N2, N3**		26 (5–30)	22.5 (12.5–35)
** *p* **		0.53	0.68
**TNM stage**			
**I**		30.5 (15–48)	10 (10–30)
**II**		40 (40–40)	20 (5–40)
**III**		10 (5–30)	20 (10–30)
**IV**		30 (26–60)	28 (15–40)
** *p* **		IA	0.35
**Histological grading**			
**G1**		22.5 (10–46)	15 (10–35)
**G2**		20.5 (5–40)	20 (10–32.5)
**G3**		30 (30–30)	22.5 (10–40)
** *p* **		IA	0.82
**Local recurrence**			
**present**		40 (20–40)	35 (20–40)
**absent**		20 (10–37.5)	20 (10–35)
** *p* **		IA	0.07
**Nodal recurrence**			
**present**		15 (10–40)	22.5 (10–30)
**absent**		30 (26–40)	20 (10–40)
** *p* **		IA	0.59

**Table 4 diagnostics-12-01059-t004:** Risk factors for premalignant laryngeal lesions (PLLs), laryngeal cancer (LC) and PLLs with progression to LC (PLL→LC), statistically significant data are in bold (*p* < 0.05).

	Univariate Analysis						Multivariate Analysis					
	PLLs		PLLs→LC		LC		PLLs chi^2^ = 35.5; *p* < 0.001		PLLs→LC Chi^2^ = 22.29 *p* = 0.0001		LC chi^2^ = 160; *p* < 0.0001	
	**OR** **(95% CI)**	*p*	OR (95% CI)	*p*	OR (95% CI)	*p*	OR (95% CI)	*p*	OR (95% CI)	*p*	OR (95% CI)	*p*
**>55 year-old**	**3.05** **(1.72–5.44)**	**<0.001**	0.52 (0.20–1.32)	0.17	**6.56** **(3.38–12.72)**	**<0.01**	**3.16** **(1.73–5.79)**	**<0.001**	0.91 (0.69–1.58)	0.89	**6.76** **(3.52–13)**	**<0.001**
**Male gender**	**2.3** **(1.36–3.89)**	**0.002**	6.69 (0.9–50)	0.065	**8.64** **(3.95–18.91)**	**<0.01**	**2.47** **(1.42–4.3)**	**0.001**	19.84 (2.03–194.09)	0.01	**8.42** **(3.9–18.19)**	**<0.001**
**Smoking**	1.42 (0.95–2.24)	0.15	1.37 (0.48–3.90)	0.54	**2.74** **(1.34–3.21)**	**<0.01**	1.18 (0.66–2.11)	0.06	0.87 (0.69–1.48)	0.98	**3.55** **(2.21–5.7)**	**<0.001**
**Alcohol intake**	0.88 (0.54–1.38)	0.97	1.86 (0.60–5.77)	0.28	**3.98** **(2.68–5.9)**	**<0.001**	0.95 (0.65–1.38)	0.79	0.90 (0.60–1.30)	0.98	1.18 (0.86–2.01)	0.15
**Primary site epiglottic**	1.28 (0.90–2.09)	0.12	1.33 (0.91–2.13)	0.14	0.93 (0.61–1.42)	0.71	0.86 (0.50–1.40)	0.98	1.76 (0.61–4.87)	0.27	1.20 (0.76–2.21)	0.23
**Cyclin D1 overexpression**	0.85 (0.51–1.28)	0.85	1.23 (0.98–1.72)	0.12	0.91 (0.61–1.38)	0.79	0.79 (0.51–1.20)	0.99	1.66 (0.66–5.07)	0.29	0.50 (0.20–1.30)	0.2
**A allele carrier**	2.11 (0.64–10.45)	0.21	2.23 (0.01–1.99)	0.28	**2.55** **(1.33–5.9)**	**0.006**	1.21 (0.69–2.22)	0.07	1.19 (0.72–2.35)	0.09	**3.2** **(1.46–7.04)**	**<0.001**
**High-grade dysplasia**	1.68 (1.52–2.31)	0.08	**1.68** **(1.7–2.6)**	**0.025**			0.88 (0.57–1.30)	1.01	0.85 (0.47–1.30)	1.02		

**Table 5 diagnostics-12-01059-t005:** Overall survival in laryngeal cancer (LC) patients (n = 101) in univariate and multivariate Cox regression analysis, differentiating point of Cyclin D1 expression group was median (chi^2^ = 47.53; *p* < 0.0001) or cut-off point nominate in ROC curves analysis at the level of 15% (chi^2^ = 48.29; *p* < 0.0001).

			HR (95%CI)	*p*
**Univariate** **Cox regression**		**Male gender**	3.08 (1.23–7.71)	0.016
		**Primary site epiglottic**	5.87 (2.89–12.01)	<0.001
		**Tumor extension T3, T4**	5.48 (1.95–15.44)	0.001
		**Nodal stage N1, N2, N3**	6.62 (3.29–13.3)	<0.001
		**Histological grading G3**	3.25 (1.93–5.48)	<0.001
		**Cyclin D1 overexpression (median)**	1.02 (1.0–1.04)	0.025
		**Cyclin D1 overexpression (ROC)**	3.03 (1.75–5.24)	<0.001
**Multivariate** **Cox regression**	**(Differentiating point—median)**	**Nodal stage N1, N2, N3**	5.4 (2.59–11.27)	<0.001
		**Histological grading G3**	2.52 (1.45–4.39)	0.001
		**Cyclin D1 overexpression (median)**	1.03 (1.00–1.05)	0.02
	**(Differentiating point—ROC)**	**Nodal stage N1, N2, N3**	5.05 (2.44–10.43)	<0.001
		**Histological grading G3**	2.5 (1.42–4.40)	0.002
		**Cyclin D1 overexpression (ROC)**	2.47 (1.16–5.25)	0.019

## Data Availability

The data presented in this study are available on request from the corresponding author (W.P.) upon reasonable request.

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
