# Peer review of "G870A Polymorphic Variants of CCND1 Gene and Cyclin D1 Protein Expression as Prognostic Markers in Laryngeal Lesions"

_diagnostics, 2022, doi:10.3390/diagnostics12051059_

Round 1
Reviewer 1 Report
Line 30 …add anatomic before word region…
Line 44 …edit a sentence by removing the end part because it is confusing…so the sentence should be…”Additionally, early diagnosis of laryngeal cancer remains a challenge.
Line 59 …of the head and neck cancers…add “the”…
Line 93 the sentence “Further aim was to identify factors that determine the course of laryngeal cancer” is to general and unnecessary either edit sentence and be more specific or remove it…
In the section Material and methods under 2.1 Study population and epidemiologic data authors did not describe what did they included in the group of “healthy controls” what type of lesions pathohistologically…
2.2. Immunohistochemistry section starts in very confusing way with thickness of sections, and you state that those are HE slides then you start describing immunohistochemistry protocol so something is very off in this section please edit the description of method…someone who is experienced with analysis workflow should check it and write it down…no need for going into details because those methods are standard of practice and in everyday use…and after in the results you mention level of cyclin d1 expression moderate, high but criteria for low, moderate or high expression of cyclin d1 are not described in the evaluation of immunohistochemistry…
In the evaluation section it would be good if you could state how big was the “examined area” or you have used 10 HPF…and usually more relevant results are obtained when in the assessment calculation percentage and intensity of staining was used but you stated that you did not see major variations in staining…is that in all squamous epithelial lesions that you have examined LC, PLL, healthy control?!?
2.3 Genotyping
Why did you perform genotyping from the whole blood instead from tissue sections that you examined, at least in the precancerous lesions and laryngeal cancer if not in healthy controls…it would give more accurate situation because you are examining changes within specific body parts, tissue that has been changing locally due to exposure to some specific risk factors followed by specific morphologic changes in the squamous epithelium…that would be more appropriate…?
Line 195 use past tense so not “consists” but rather “consisted”
Line 197 sentence Patients with PLLs were observed…after their radical removal. -this sentence doesn’t make sense, edit sentence be clear of what do you want to state in the sentence…
Line 204 sentence that starts in this line “Time off from…again very confusing so please revise
Line 215 remove typesetting…you have forgot to delete it…
Section 3.3 Cyclin D1 expression in patients with laryngeal lesions-in this section in one instance you mention “initial concentrations of Cyclin D1”…as you mentioned you were examining the level of protein expression not protein concentration, immunohistochemistry can not give you that answer…very thoroughly check this section…
Figure 1. in the explanatory notes you keep mentioning terms like moderate expression, high expression of cyclin d1…but you have not specified in the Materials and methods what are the cut off for these groups, everything must be uniformly written and compared…
3.4 Analysis of risk factors for laryngeal lesions development…
Data described in this section do not give nothing new, all that has been said is already known…
Line 283 …G3 grade in NOT low histological grade…please get in order facts…and this is seen further down in the manuscript…
In all tables the examined parameters were written in small letters, it is common to start word with capital letter…
There is much literature and studies regarding cyclin D1 gene G870A polymorphism as well as on cyclin D1 expression in squamous epithelium so written results are much expected on what can be found in the literature, so I am not clear that this adds much to the existing literature on this subject, therefore manuscript either requires rewriting or additional experimental data…
Author Response
Dear Reviewer,
we appreciate the effort that you have provided for the feedback on our manuscript. Thank you for the comments and valuable improvements to our article. The changes are highlighted within the manuscript (in red), for a point-by-point response to the comments and concerns. All page numbers refer to the revised manuscript file with tracked changes.
Reviewer: Line 30 …add anatomic before word region…
Response: Thank you for the suggestion. We have decided to edit the sentence.
Reviewer: Line 44 …edit a sentence by removing the end part because it is confusing…so the sentence should be…”Additionally, early diagnosis of laryngeal cancer remains a challenge.
Response: Accordingly to your suggestion we have decided to remove the end of the sentence.
Reviewer: Line 59 …of the head and neck cancers…add “the”…
Response: We agree with this and have incorporated your suggestion in the sentence.
Reviewer: Line 93 the sentence “Further aim was to identify factors that determine the course of laryngeal cancer” is to general and unnecessary either edit sentence and be more specific or remove it…
Response: Thank you for the suggestion. Accordingly we have decided to remove the sentence.
Reviewer: In the section Material and methods under 2.1 Study population and epidemiologic data authors did not describe what did they included in the group of “healthy controls” what type of lesions pathohistologically…
Response: This is an excellent suggestion. Our control group consisted of patients operated for non-oncological purposes without previous cancer history or autoimmune disorders, which we highlight in the text. (page 3 line 114) We understand the need to mention precisely the histological lesion types. However, from control patients we collected only blood samples for genotyping, without other tissues to avoid unnecessary traumatization. In the text we added proper explanation and list of diseases of the control group. (page 3 line 118)
Added part of the sentence: septoplasty, ossiculoplasty, myringoplasty, trauma injuries (page 3 line 118)
Reviewer: 2.2. Immunohistochemistry section starts in very confusing way with thickness of sections, and you state that those are HE slides then you start describing immunohistochemistry protocol so something is very off in this section please edit the description of method…someone who is experienced with analysis workflow should check it and write it down…no need for going into details because those methods are standard of practice and in everyday use…. and after in the results you mention level of cyclin d1 expression moderate, high but criteria for low, moderate or high expression of cyclin d1 are not described in the evaluation of immunohistochemistry…
In the evaluation section it would be good if you could state how big was the “examined area” or you have used 10 HPF…and usually more relevant results are obtained when in the assessment calculation percentage and intensity of staining was used but you stated that you did not see major variations in staining…is that in all squamous epithelial lesions that you have examined LC, PLL, healthy control?!?
Response: Thank you for the suggestion. Accordingly to the reviewer’s assessment, we have decided to revised the section 2.2.
Moreover, we found the Figure 1 description to be inadequate and misleading. We changed the description accordingly to clarify the issue of low, moderate or high expression of Cyclin D1.
Revised section 2.2 Immunohistochemistry
“The appropriate tissue samples were selected from laryngeal squamous cell carcinoma and from premalignant laryngeal lesions independently. Tissue samples were selected from paraffin-embedded blocks and cut into 4-m thick sections. They were deparaffinized in xylene and dehydrated through graded alcohols. Endogenous peroxidase was blocked with 3% H2O2 after 5 minutes incubation (pressure cooking) in 10 mmol/L of citrate buffer (pH 6.0) or EDTA buffer (pH 9.0). The next step was 20 minutes incubation with the serum blocking solution. The real EnVision detection system (DAKO) was used for immunostaining (according to the manufacturer’s procedure). The mouse anti-Cyclin D1 monoclonal antibody (NCL-Cyclin D1, clone designation DCS-6, Novocastra Laboratories Ltd) was used to detect total Cyclin D1. Next 2 hours incubation of primary antibody at room temperature, then washing (3 times) with phosphate-buffered saline, 30 minutes incubation using labeled polymer-horseradish peroxidase (HRP) antimouse/rabbit Ig. The diaminobenzidine chromogen solution was added after incubation. Counterstaining was conducted with hematoxylin. Negative controls were performed by omitting the primary antibody. The nuclear staining was considered positive, regardless of cytoplasmic staining. The morphometric evaluation was made in the light microscope equipped with special program (Nikon, magnification 200 and 400x). Positive reaction was recognized as brown coloration of the cell nuclei and determined from the staining intensity as well as the percentage of immunoreactive cells in the hot spots (most highly stained area of each slide) based on a previously reported method (the index of positive cells). The intensity score was graded as 0 (no staining), 1 (weak staining), 2 (moderate staining), and 3 (severe staining).
Also, in PLLs, a quantitative indication of staining was determined by calculating the percentage of immunopositive epithelial cells to the total epithelial cells, to create an expression index (EI). Sections of the cervix with different grades of intraepithelial neoplasia were used as positive controls. Staining intensity was not included in the assessment, as only relatively minor variations in staining intensity were observed.
The results were independently assessed by specialists in histopathology. Cases with divergent results were re-evaluated simultaneously and a final opinion was made and based on consensus by all two investigators.“
New description of the Figure 1:
“Immunohistochemical expression of Cyclin D1 in:
a) PLL – visible mostly in the nuclei of basal layer cells of hyperplastic epithelium (index: 32%, magnification 200x);
b) LC: b1) in the GG homozygote patient - visible in single cancer cells (index: 5%, magnification 200x); b2) in the AA homozygote patient visible as small cellular granules in nuclei of cells (index: 41%, magnification 200x);”
Reviewer: 2.3 Genotyping
Why did you perform genotyping from the whole blood instead from tissue sections that you examined, at least in the precancerous lesions and laryngeal cancer if not in healthy controls…it would give more accurate situation because you are examining changes within specific body parts, tissue that has been changing locally due to exposure to some specific risk factors followed by specific morphologic changes in the squamous epithelium…that would be more appropriate…?
Response:
Thank you for pointing this out. There were several reasons why we used blood samples for the genotyping.
- the material was more accessible and easier to obtain both in the control and the study group.
- for the controls the blood DNA was used from a banked set which made it impossible to extract tissues from these individuals but it optimized the need for additional collections.
- when isolating from formalin-fixed tissues, the costs are higher, the procedure is more laborious and the quality of the isolates is lower.
- in the neoplastic tissue there may be genetic changes that are not constitutional but somatic changes, we were assessing the patients’ genome
- the aim of our study did not include examination of changes in genotypes distribution between cancer and healthy tissues. Furthermore, cancer tissue is combined from healthy and cancer cells, which makes that exact genotyping for the healthy and cancer tissue technically impossible.
Therefore, we found the genotyping based on DNA isolated from blood as much easier and more reliable in our study.
Reviewer: Line 195 use past tense so not “consists” but rather “consisted”
Response: We agree with this and have incorporated your suggestion in the manuscript.
Reviewer: Line 197 sentence Patients with PLLs were observed…after their radical removal. -this sentence doesn’t make sense, edit sentence be clear of what do you want to state in the sentence…
Response: We think this is an excellent suggestion. We have decided to edit the sentence.
Changed sentence: After the surgery, patients with PLLs were observed up to 15.3 years (mean 13.1, SD 2.2; median 13.0). (page 5 line 221)
Reviewer: Line 204 sentence that starts in this line “Time off from…again very confusing so please revise
Response: Thank you for pointing this out. We have changed the sentence.
Changed sentence: “In 27 patients with laryngeal cancer (26.7%) we observed local or nodal recurrences, which appeared in a time range between 12 months to 8.7 years (mean 23; BC 25.13; median 13).” (page 5 line 226)
Reviewer: Line 215 remove typesetting…you have forgot to delete it…
Response: We agree with this and have incorporated your suggestion in the manuscript.
Reviewer: Section 3.3 Cyclin D1 expression in patients with laryngeal lesions-in this section in one instance you mention “initial concentrations of Cyclin D1”…as you mentioned you were examining the level of protein expression not protein concentration, immunohistochemistry cannot give you that answer…very thoroughly check this section…
Response: Thank you for pointing this out. We were examining the level of protein expression, and we used the statement “concentration” wrongly as an translation error. Therefore, we have changed the word “concentrations” for “protein tissue levels”.
Reviewer: Figure 1. in the explanatory notes you keep mentioning terms like moderate expression, high expression of cyclin d1…but you have not specified in the Materials and methods what are the cut off for these groups, everything must be uniformly written and compared…
Response: Thank you for your suggestion. According to your comment we decide to change the confusing description for Figure 1.
New description of the Figure 1.
“Immunohistochemical expression of Cyclin D1 in:
a) PLL – visible mostly in the nuclei of basal layer cells of hyperplastic epithelium (index: 32%, magnification 200x);
b) LC: b1) in the GG homozygote patient - visible in single cancer cells (index: 5%, magnification 200x); b2) in the AA homozygote patient visible as small cellular granules in nuclei of cells (index: 41%, magnification 200x);”
Reviewer: 3.4 Analysis of risk factors for laryngeal lesions development…
Data described in this section do not give nothing new, all that has been said is already known…
Response: We agree with the reviewer’s assessment. We decided to include this section because our primary intention was to include in the manuscript all obtained results, even though not all of them give more than what is already known. If it will be clearer without this section, we will consider removing it and include in the data as a supplementary material.
Reviewer: Line 283 …G3 grade in NOT low histological grade…please get in order facts…and this is seen further down in the manuscript…
Response: We agree with the reviewer’s assessment. We write “low” wrongly as a translation error. We have changed sentences throughout the manuscript. (page 9 line 401, 409, 417)
Reviewer: In all tables the examined parameters were written in small letters, it is common to start word with capital letter…
Response: We agree with this and have incorporated your suggestion throughout the manuscript.
Reviewer: There is much literature and studies regarding cyclin D1 gene G870A polymorphism as well as on cyclin D1 expression in squamous epithelium so written results are much expected on what can be found in the literature, so I am not clear that this adds much to the existing literature on this subject, therefore manuscript either requires rewriting or additional experimental data…
Response: While we appreciate the reviewer’s feedback, we respectfully disagree. We think this study makes a valuable contribution to the field because it comprehensively explores Cyclin D1 protein expression and G870A polymorphism in patients with laryngeal cancer and premalignant laryngeal lesions including the follow-up up to the 15 years, giving to the reader the overall look on the field.
Moreover, not all results in the literature are consistent so we find it interesting to perform a study that relates to them as much as possible.
Reviewer 2 Report
The paper on G870A polymorphic variants of CCND1 gene and Cyclin D1 protein expression as prognostic markers in laryngeal lesions is presented. It is an interesting contribution to the field, especially in the context of comprehensively addressing the association of CCND1 polymorphism and Cyclin D1 protein expression with risk for both laryngeal precancerous lesions and laryngeal carcinoma. Also, a considerably long follow-up period allowed the conclusions regarding progression from precancerous lesions to laryngeal carcinoma.
Generally, the style is clear and concise, the usage of English language is appropriate, making the text easy to follow. Statistical analyses are adequate, providing a good basis for the conclusions. The list of references is appropriate.
I have only some minor remarks:
- the PLL group included PLLs with progression to cancer (n=17). At which time point(s) have samples analysed for Cyclin D1 expression been taken in this subgroup of PLL patients? Into which group you put their samples when their progressed to cancer – still PLL or LC group? Please, add the information to the text.
- it seems rather strange that staining intensity was not included in the assessment, and that only relatively minor variations in staining intensity were observed, since there are papers describing significant differences in staining intensity and histological distribution of Cyclin D1 expression in laryngeal precancerous lesions and laryngeal carcinoma (for example, Marzic D. et al. The expression of ribonuclear protein IMP3 in laryngeal carcinogenesis. Pathol Res Pract. 2020 Jun;216(6):152974. doi: 10.1016/j.prp.2020.152974.) Also, differences in staining intensity can be observed on the very figures the authors provided in their paper. Please, clarify.
- regarding tobacco exposure, you mention the number of “cigarettes per year”, but at the same time only two categories: non-smokers and smokers. What about passive smokers? Did you consider them? Please clarify and add the appropriate information and explanation to the text.
- the part of the discussion dealing with the expression of Cyclin D1 protein seems biased, since you only mention the studies which are in line with your findings, while there are studies showing different results (also please see Marzic D. et al. The expression of ribonuclear protein IMP3 in laryngeal carcinogenesis. Pathol Res Pract. 2020 Jun;216(6):152974. doi: 10.1016/j.prp.2020.152974.). Please rewrite this part of the discussion.
Author Response
Dear Reviewer,
we appreciate the effort that you have provided for the feedback on our manuscript. Thank you for your favorable assessment of our work - the comments and valuable improvements. The changes are highlighted within the manuscript (in red), for a point-by-point response to the comments and concerns. All page numbers refer to the revised manuscript file with tracked changes.
Reviewer: The paper on G870A polymorphic variants of CCND1 gene and Cyclin D1 protein expression as prognostic markers in laryngeal lesions is presented. It is an interesting contribution to the field, especially in the context of comprehensively addressing the association of CCND1 polymorphism and Cyclin D1 protein expression with risk for both laryngeal precancerous lesions and laryngeal carcinoma. Also, a considerably long follow-up period allowed the conclusions regarding progression from precancerous lesions to laryngeal carcinoma.
Generally, the style is clear and concise, the usage of English language is appropriate, making the text easy to follow. Statistical analyses are adequate, providing a good basis for the conclusions. The list of references is appropriate.
I have only some minor remarks: the PLL group included PLLs with progression to cancer (n=17). At which time point(s) have samples analysed for Cyclin D1 expression been taken in this subgroup of PLL patients? Into which group you put their samples when their progressed to cancer – still PLL or LC group? Please, add the information to the text.
Response:
We agree with the reviewer that this is very important information that is not clear for the reader. We have incorporated your suggestion in the methods section. We collect the tissue samples only once at the beginning of the observation. We separate the PLLs-> LC subgroup to explore the differences between premalignant lesions that are likely to progress to cancer from those that remain precancerous in a long period.
Added sentences: “The samples from the PLL with progression to LC group were taken and analyzed at the beginning when they were diagnosed as the premalignant lesions. We do not collect another samples when the lesion progressed to cancer during the follow-up.” (page 3 line 112-115)
Reviewer: - it seems rather strange that staining intensity was not included in the assessment, and that only relatively minor variations in staining intensity were observed, since there are papers describing significant differences in staining intensity and histological distribution of Cyclin D1 expression in laryngeal precancerous lesions and laryngeal carcinoma (for example, Marzic D. et al. The expression of ribonuclear protein IMP3 in laryngeal carcinogenesis. Pathol Res Pract. 2020 Jun;216(6):152974. doi: 10.1016/j.prp.2020.152974.) Also, differences in staining intensity can be observed on the very figures the authors provided in their paper. Please, clarify.
Response: Thank you for pointing out. Accordingly to your suggestion we decided to revised the section 2.2 and the Figure 1 description.
Revised section 2.2 Immunohistochemistry
“The appropriate tissue samples were selected from laryngeal squamous cell carcinoma and from premalignant laryngeal lesions independently. Tissue samples were selected from paraffin-embedded blocks and cut into 4-m thick sections. They were deparaffinized in xylene and dehydrated through graded alcohols. Endogenous peroxidase was blocked with 3% H2O2 after 5 minutes incubation (pressure cooking) in 10 mmol/L of citrate buffer (pH 6.0) or EDTA buffer (pH 9.0). The next step was 20 minutes incubation with the serum blocking solution. The real EnVision detection system (DAKO) was used for immunostaining (according to the manufacturer’s procedure). The mouse anti-Cyclin D1 monoclonal antibody (NCL-Cyclin D1, clone designation DCS-6, Novocastra Laboratories Ltd) was used to detect total Cyclin D1. Next 2 hours incubation of primary antibody at room temperature, then washing (3 times) with phosphate-buffered saline, 30 minutes incubation using labeled polymer-horseradish peroxidase (HRP) antimouse/rabbit Ig. The diaminobenzidine chromogen solution was added after incubation. Counterstaining was conducted with hematoxylin. Negative controls were performed by omitting the primary antibody. The nuclear staining was considered positive, regardless of cytoplasmic staining. The morphometric evaluation was made in the light microscope equipped with special program (Nikon, magnification 200 and 400x). Positive reaction was recognized as brown coloration of the cell nuclei and determined from the staining intensity as well as the percentage of immunoreactive cells in the hot spots (most highly stained area of each slide) based on a previously reported method (the index of positive cells). The intensity score was graded as 0 (no staining), 1 (weak staining), 2 (moderate staining), and 3 (severe staining).
Also, in PLLs, a quantitative indication of staining was determined by calculating the percentage of immunopositive epithelial cells to the total epithelial cells, to create an expression index (EI). Sections of the cervix with different grades of intraepithelial neoplasia were used as positive controls. Staining intensity was not included in the assessment, as only relatively minor variations in staining intensity were observed.
The results were independently assessed by specialists in histopathology. Cases with divergent results were re-evaluated simultaneously and a final opinion was made and based on consensus by all two investigators.”
New description of the Figure 1:
“Immunohistochemical expression of Cyclin D1 in:
- a) PLL – visible mostly in the nuclei of basal layer cells of hyperplastic epithelium (index: 32%, orig. magn. 200x);
- b) LC: b1) in the GG homozygote patient - visible in single cancer cells (index: 5%, orig. magn. 200x); b2) in the AA homozygote patient visible as small cellular granules in nuclei of cells (index: 41%, orig. magn. 200x); “
Reviewer: - regarding tobacco exposure, you mention the number of “cigarettes per year”, but at the same time only two categories: non-smokers and smokers. What about passive smokers? Did you consider them? Please clarify and add the appropriate information and explanation to the text.
Response: Thank for your suggestion. We decided to divide patients into smokers and non-smokers, according to their daily habits. In the group of smokers, we included current and former smokers, although the number of the latter was traceable and the cessation took place over a period of less than one decade, so we did not include this detailed information in the text. Smokers were then divided into light and heavy smokers due to the number of the cigarettes per year respectively below and above the median (7300 cigarettes per year). (page 3 line 123) We did not collect the data about passive smoking; however you point out interesting issue for further studies. We agree that this is a potential limitation of the study.
Accordingly, we decide to mention in the text that we only assessed the habits, not the passive smoking.
Added sentence: “We assessed only the data about active smoking” (page 3 line 124)
Reviewer: - the part of the discussion dealing with the expression of Cyclin D1 protein seems biased, since you only mention the studies which are in line with your findings, while there are studies showing different results (also please see Marzic D. et al. The expression of ribonuclear protein IMP3 in laryngeal carcinogenesis. Pathol Res Pract. 2020 Jun;216(6):152974. doi: 10.1016/j.prp.2020.152974.). Please rewrite this part of the discussion.
Response:
Revised part of the discussion:
“Moreover, in our study, among patients with premalignant laryngeal lesions, A allele carriers of the CCND1 gene G870A polymorphism had higher expression of Cyclin D1 protein compared to GG homozygotes. It was observed in patients with PLLs which progressed to cancer as well. This trend was also visible in high-grade dysplasia cases compared to the patients with early stages of PLLs. Our results showed that expression of Cyclin D1 increased on progression along with the lesion - from dysplasia (10-15%) to carcinoma (30%). Marzic et al obtained similar pattern of expression but with different values – 35% for premalignant lesions and 45% for cancer. [42] These results suggests that cyclin D1 expression can be used for differentiating the PLLs from cancer but not from the healthy tissue. However, some researchers showed that Cyclin D1 appeared to be highly sensitive (81.2%) and specific marker (83.9%) not only in differentiating LC from PLLs but also from healthy laryngeal mucosa (sensitivity: 81.2%; specificity: 41.4%) [43] Recent studies revealed also that the expression of Cyclin D1 protein was correlated with tumor stage, metabolic volume of tumor, glycolysis in le-sion and maximum standardized uptake value (SUV max) in patients diagnosed with laryngeal cancer; and was suggested as diagnostic and follow-up marker. [34, 44] Moreover studies showed that overexpression of Cyclin D1 is associated with lymph node metastases (OR 2.26; 95 % CI 1.61–3.16). [45] We did not observe the association with tumor extension, nodal involvement, TNM stage and histological grading of LC. Our results show that Cyclin D1 protein expression was significantly associated with the shorter survival time in laryngeal cancer patients, which is similar to the results of other studies. [46]”
Round 2
Reviewer 1 Report
Page 3 lines 117-119...so as you wrote in the your explanatory notes write it in the text of the manuscript that from healthy controls you collected only blood...because it is still not clear to the readers...to avoid misunderstanding...
Everything else has been corrected as suggested…
Author Response
Dear Reviewer,
thank you for your assessment of our work. We believe that thanks to your comments our article has become more understandable for the readers.
The change is highlighted within the manuscript (in red).
Reviewer: Page 3 lines 117-119...so as you wrote in the your explanatory notes write it in the text of the manuscript that from healthy controls you collected only blood...because it is still not clear to the readers...to avoid misunderstanding...
Everything else has been corrected as suggested…
Response: Thank you for your suggestion. We have added the sentence in the text.
Added sentence: “Only blood samples for the genotyping were obtained from controls” (page: 3 line 117)